# The Use of Lignin as a Microbial Carrier in the Co-Digestion of Cheese and Wafer Waste

**DOI:** 10.3390/polym11122073

**Published:** 2019-12-12

**Authors:** Agnieszka A. Pilarska, Agnieszka Wolna-Maruwka, Krzysztof Pilarski, Damian Janczak, Krzysztof Przybył, Marzena Gawrysiak-Witulska

**Affiliations:** 1Institute of Food Technology of Plant Origin, Poznań University of Life Sciences, Wojska Polskiego 31, 60-637 Poznań, Poland; kprzybyl@up.poznan.pl (K.P.); marzena.gawrysiak-witulska@up.poznan.pl (M.G.-W.); 2Department of General and Environmental Microbiology, Poznań University of Life Sciences, Wojska Polskiego 31, 60-637 Poznań, Poland; amaruwka@up.poznan.pl; 3Institute of Biosystems Engineering, Poznań University of Life Sciences, Wojska Polskiego 50, 60-637 Poznań, Poland; pilarski@up.poznan.pl (K.P.); djanczak@up.poznan.pl (D.J.)

**Keywords:** lignin, microbial carrier, cheese waste, wafer waste, biogas production

## Abstract

The aim of the article was to present the effects of lignin grafted with polyvinylpyrrolidone (PVP) as a microbial carrier in anaerobic co-digestion (AcoD) of cheese (CE) and wafer waste (WF). Individual samples of waste cheese and wafers were also tested. The PVP modifier was used to improve the adhesive properties of the carrier surface. Lignin is a natural biopolymer which exhibits all the properties of a good carrier, including nontoxicity, biocompatibility, porosity, and thermal stability. Moreover, the analysis of the zeta potential of lignin and lignin combined with PVP showed their high electrokinetic stability within a wide pH range, that is, 4–11. The AcoD process was conducted under mesophilic conditions in a laboratory by means of anaerobic batch reactors. Monitoring with two standard parameters: pH and the VFA/TA ratio (volatile fatty acids-to-total alkalinity ratio) proved that the process was stable in all the samples tested. The high share of N–NH_4_^+^ in TKN (total Kjeldahl nitrogen), which exceeded 90% for WF+CE and CE at the last phases of the process, proved the effective conversion of nitrogen forms. The microbiological analyses showed that eubacteria proliferated intensively and the dehydrogenase activity increased in the samples containing the carrier, especially in the system with two co-substrates (WF+CE/lignin) and in the waste cheese sample (CE/lignin). The biogas production increased from 1102.00 m^3^ Mg^−1^ VS (volatile solids) to 1257.38 m^3^ Mg^−1^ VS in the WF+CE/lignin sample, and from 881.26 m^3^ Mg^−1^ VS to 989.65 m^3^ Mg^−1^ VS in the CE/lignin sample. The research results showed that the cell immobilization on lignin had very positive effect on the anaerobic digestion process.

## 1. Introduction

The dairy industry generates significant amounts of waste at various stages of the production chain, especially in cheese production processes [1]. In spite of the specific possibilities to use selected types of waste, including permeate, whey protein concentrate, and isolate products [2], other liquid and solid wastes produced in large amounts require effective disposal methods.

At present, anaerobic digestion (AD) is thought to be the most sustainable solution in waste management and energy recovery, also in the dairy sector [3]. Although this is an environment-friendly and promising method, its implementation may cause problems, mostly due to the chemical composition of dairy waste (i.e., acidic pH, high concentration of mineral salts) [4]. The breakdown of typical protein waste containing casein involves the release of ammonia and hydrogen sulphide, which inhibit the anaerobic digestion [3]. In this case, the accumulation of volatile fatty acids (VFA) originating from the coexistent fat may occur equally frequently. In order to solve technical problems, various configurations of reactors, operating under mesophilic or thermophilic conditions, were tested [5,6,7]. There have also been attempts to stabilize the system by supplementation with alkalizing compounds [8]. However, for economic reasons it is more common to combine dairy waste with other co-substrates, buffering the system and improving the balance of carbon and nutrients [9,10,11]. The composition of an effective co-substrate should complement the basic substrate and cause synergistic effects [12]. Confectionery waste is the material that potentially meets the abovementioned conditions regarding cheese waste. The confectionery industry continuously generates large amounts of waste. Apart from being recycled, it should be disposed of or used in accordance with the principles of sustainable development [13]. As stated in previous studies, confectionery waste undoubtedly has high biochemical methane potential (BMP) due to its composition and physicochemical properties [14,15].

Researchers try to modify the AD process conditions to ensure better distribution of microbial species in a suitable habitat, provide them with constant access to valuable nutrients and appropriate pH in their environment so as to improve their metabolic activity [16]. One of the methods improving the condition of methanogens and other mixed cultures is their immobilization with an appropriate carrier [17]. Interactions between microorganisms and the carrier result in the formation of a compact and stable biofilm, which naturally becomes a more active catalyst of bioconversion processes. Cells have a natural tendency to adhere to surfaces. This feature is of key significance in the cell immobilization method. Its driving force is the deficit of nutrients in the environment. The concentration of nutrients is usually slightly higher near the surface [18]. The choice of an adequate carrier which matches the conditions of the process is decisive to its success. So far, various materials have been tested as carriers in the methane fermentation process, for example, artificial polymers, activated carbon, pine shavings, or natural zeolites [19]. As results from the reference publications, the carrier limited the release and the effect of inhibitors (e.g., ammonia), desensitized cells to changes in the environment (pH, temperature, access to the medium), stabilized the system, and increased the bioconversion of materials.

Lignin is one of the most common renewable substances after cellulose. It is a complex aromatic biopolymer mainly composed of p-coumaryl, coniferyl, and sinapyl alcohols—three fundamental monomers differing in the degree of methoxylation [20]. Due to the valuable properties of the compound, such as porosity, well-developed specific surface area, exceptional absorbency, thermoplasticity, bioactivity, and specific structure, it has been used for various innovative purposes [21,22]. In addition, lignin has been proved to activate pancreatic α-amylase and lipase, where the immobilized enzymes achieved high activity and stability [23]. According to reports, this material has also been used for the synthesis of carrier composites with chitin and silica [24].

In view of the findings listed above and the fact that lignin meets all the requirements of a carrier in the AD process (above all, porosity, but also nontoxicity, biocompatibility, thermal and electrokinetic stability, resistance to hydrolytic enzymes), Pilarska et al. (2018, 2019) tested this biopolymer in the anaerobic breakdown of selected organic waste as a lignin/polyvinylpyrrolidone (PVP) system [25,26]. The author found that it may also be important that lignin is capable of adsorbing heavy metals and other chemical compounds that are potential inhibitors of the AD of sewage sludge [27,28]. Polyvinylpyrrolidone (PVP) is a linear water-soluble polymer, whereas in the cross-linked version it is insoluble, has excellent wetting properties, readily forms biofilms, and is often used in microbiology [29]. The task of PVP is to improve cell adhesion to the carrier surface.

The goal of the study was to verify the amount of biogas/methane produced in anaerobic co-digestion of waste cheese (CE) and waste wafers (WF) with the addition of lignin combined with polyvinylpyrrolidone (−[CH_2_CH(C_4_H_6_NO)]_n_−, PVP), and without the carrier—as the control variant. The process was conducted in a laboratory under mesophilic conditions in anaerobic batch digestion reactors. Stabilized sewage sludge was used as the inoculum. During the experiment, the fermenting suspension was analyzed for pH, volatile fatty acids/total alkalinity (VFA/TA), total Kjeldahl nitrogen (TKN), ammonium nitrogen (N–NH_4_^+^), chemical oxygen demand (COD), and bacterial counts and enzymatic activity.

## 2. Materials and Methods

### 2.1. Feedstock and Inoculum

Waste cottage cheese (CE) from a dairy and waste wafers (WF) from a confectionery near Poznań were used as substrates. Stabilized sewage sludge used in the experiment as an inoculum was acquired from the Poznań Central Sewage Treatment Plant (Poznań, Poland). Kraft lignin, which was used as a carrier, and polyvinylpyrrolidone—a carrier modifier, were purchased from Sigma-Aldrich, Steinheim am Albuch, Germany. The physicochemical properties of the waste materials used in the experiment are shown in Table 1.

### 2.2. Experimental Procedure

#### 2.2.1. Sample Preparation

The following control samples and samples with a carrier were tested: WF, WF/lignin, CE, CE/lignin, WF+CE, WF+CE/lignin, inoculum, inoculum/lignin. The content of substrates and inoculum in the batches followed the standard [30] with guidelines for the digestion of organic materials. As recommended, the content of dry organic matter in the inoculum ranged from 1.5% to 2%, whereas the content of dry matter in the batch did not exceed 10% [31]. Table 2 shows the composition of the batches and their most important physicochemical parameters.

#### 2.2.2. Carrier Preparation

The lignin+PVP cell carrier was prepared by making a wet mechanical connection (with 50 mL H_2_O) of 20 g lignin and 3.5 g polyvinylpyrrolidone. The carrier was applied to appropriate substrate/inoculum batches (see Table 2), and then the whole was mixed vigorously. The carrier was added to the mixture at an amount based on data in reference publications [32,33]. The authors of this study made partial physicochemical analysis of the lignin+PVP material in earlier tests [25,26], which showed that it met essential requirements for the microbial carrier in the AD process.

#### 2.2.3. Anaerobic Digester Setup

The anaerobic digestion process was conducted in a multi-chamber bioreactor, shown in Figure 1. In total, there were 24 digestion chambers in this experiment (each sample was tested in triplicate). The capacity of the bioreactors was 1.4 L (5). They were filled with the batch up to 1 L, which was mixed once a day. The bioreactors were located in a water jacket (4) connected to a heater (1), thanks to which the process could be conducted within the set temperature range (mesophilic conditions) [14,34]. The biogas produced in the process flowed (7) to the tanks (8) freely, where it was stored. The hydraulic retention time (HRT) was 21 days. According to the standard [35], the experiment was continued until the daily biogas production in all the bioreactors dropped below 1% of the total amount of biogas produced.

#### 2.2.4. Qualitative and Quantitative Analysis of Biogas

The volume of biogas produced was measured every 24 h. The quantitative composition of biogas was measured with Geotech GA5000 gas instrument [31].

The efficiency of biogas production (m^3^ Mg^−1^) from dry matter or dry organic matter was estimated according to results of the experiment. The biogas efficiency for individual substrates was calculated by subtraction of the volume of biogas produced from the inoculum alone from the volume of biogas produced from the samples. The equation presented by the authors in earlier studies was used to calculate the cumulative amount of biogas generated from the inoculum (digested sewage sludge) in the bioreactors with a given substrate or mixture of substrates [25,26].

### 2.3. Analytical Techniques

#### 2.3.1. Physicochemical Analysis

The pH (potentiometric analysis) and electrolytic conductivity of the substrates and batches were analyzed with an Elmetron CP-215. The content of total solids (TS) was measured by drying at 105 °C (Zalmed SML dryer, Zalmed, Łomianki, Poland). The content of volatile solids (VS) was measured by combustion at 550 °C (MS Spectrum PAF 110/6 furnace, MS Spectrum, Warszawa, Poland) [25].

The substrates and the digested materials were analyzed using the following methods and devices: the carbon content—combustion at 900 °C, CO_2_–OI Analytical analyzer; nitrogen—titration, Kjeldahl method with 0.1 n HCl; ammonium nitrogen—distillation and titration with 0.1 n HCl; phosphorus—the mineralization of phosphorus compounds with nitric acid (microwave furnace, Milestone), then by spectrophotometric analysis (Varian Cary 50); chemical oxygen demand (CODr)—titration, dichromate method (potassium dichromate, concentrated sulphuric acid, silver sulphate as catalyst); light metal ions—inductively coupled plasma optical emission spectrometry (ICP-OES, Sysmex, Hoeilaart, Belgium), JY 2000 2 ICP-OES spectrometer (ISA Jobin Yvon, Edison, NJ, USA) [15,36].

The content of VFA (volatile fatty acids), TA (total alkalinity), and the VFA/TA ratio (volatile fatty acids-to-total alkalinity ratio) in the fermenting batch was measured by collecting a 5 mL sample (5 mL of the fermentation substrate). Next, it was titrated with 0.1 N of sulphuric acid solution (H_2_SO_4_) up to pH 5.0 to calculate the TA value. The VFA value was measured after the second titration step between pH 5.0 and pH 4.4 [15].

The porous structure parameters of the lignin and lignin+PVP material were measured using an ASAP 2420 instrument (Micromeritics Instrument Co., Norcross, GA, USA). Before performing the analysis, the samples were degassed in vacuum at a temperature of 90 °C. The BJH (Barrett–Joyner–Halenda) method was applied to measure the total volume and pore size distribution [25].

Zeta potential values of the kraft lignin and lignin+PVP carrier were measured with a Zetasizer Nano ZS apparatus equipped with an autotitrator (Malvern Instruments Ltd., Cambridge, UK). It was possible to make measurements by combining electrophoresis and laser measurement of particle mobility based on the Doppler effect. In order to measure the electrophoretic mobility, a sample (0.01 g) was suspended in an electrolyte solution with variable ionic strength (25 mL), which was 0.001 M NaCl. Next, the suspension was placed in the autotitrator vessel. The entire system was titrated with 0.2 M HCl and 0.2 M NaOH solutions. pH was controlled by means of a glass electrode [24].

#### 2.3.2. Microbial Analysis

The total count of bacteria in six materials collected during the digestion process was measured with the direct method under a fluorescent microscope (Zeiss), using modified fluorescent in situ hybridization (FISH) according to [37]. Microbiological analyses were made at four terms: I—the 2nd day of the experiment, II—the 4th day of the experiment, III—the 8th day of the experiment, IV—the 14th day of the experiment, V—the 18th day of the experiment, VI—the 20th day of the experiment. 

The fermented sludge (0.01 mL) was placed on the surface of slides using a Breed chamber. Next, it was fixed with a 4% PFA solution (paraformaldehyde). After that, the materials were washed in a phosphate buffer solution (PBS) three times and 0.5% Triton solution was added. Then, the samples were washed with the PBS three times again. Next, they were placed in an alcohol series (70%, 80%, 96%). After adding a 70% formamide solution, the genetic probe EUB338 GCT GCC TCC CGT AGG AGT [38] was applied at a concentration of 25 ng µL^−1^. It was stained with a Cy5 fluorescent dye, suspended in a solution which consisted of: 5M NaCl, 1M Tris/HCl, 25% formamide, 10% SDS, and ddH2O. The digestate samples were incubated for 24 h in darkness, at 37 °C. Next, the preparations were analyzed with a fluorescence microscope equipped with an AxioCam MRc5 color digital camera. The image was analyzed by means of the AxioVision 4.8 software (AxioVision LE 4.8 2.0, Carl Zeiss Microscopy, LLC, White Plains, NY, USA) [25]. 

At the last term of analyses (VI), the count of methane microorganisms of the *Archaea* domain was measured using the fluorescence in situ hybridization method. The microorganisms in the fermented waste samples were detected with the ARCH915 GTG CTC CCC CGC CAA TTC CT probe, proposed by [39], stained with the Cy5 dye, which was used to detect these microorganisms in the fermented waste samples under analysis.

#### 2.3.3. Biochemical Analysis

The samples were analyzed biochemically using the spectrophotometric method. The dehydrogenase activity (DHA) was measured with the method developed by [40], with some modifications. The samples (5 mL) were incubated for 24 h with 2,3,5-triphenyltetrazolium chloride (TTC) at 30 °C, pH 7.4. Triphenylformazan (TPF) was produced, extracted with 96% ethanol, and measured spectrophotometrically at 485 nm. The dehydrogenase activity was presented in units μmol TPF g-1 DM of waste 24 h^−1^ [41].

#### 2.3.4. Statistical Analysis

The Statistica 12.0 program (Statistica, formerly-StatSoft Inc., Oklahoma, OK, USA) was used for statistical analyses. Two-way analysis of variance was used to determine the significance of variation in the count and activity of the microorganisms, depending on the experimental variant and the term of the analysis.

A stepwise regression procedure descending from the fifth grade was used to present the results of microbial and enzymatic analyses so as to select the multiple regression model. The coefficient of determination R^2^ was used to fit the model. Pearson’s linear correlation coefficient was calculated to identify the type of dependence between the count and activity of microorganisms and the chemical parameters of waste fermented.

## 3. Results and Discussion

### 3.1. Physicochemical Properties of Substrates

The waste cheese (CE) used in the experiment was primarily a source of organic matter and macroelements, as indicated by VS (95.75 wt%_TS_) and chemical oxygen demand (86.58 mg L^−1^)—see Table 1. What might be a matter of concern with this type of waste is its high content of calcium (849 mg kg^−1^) and sodium (433 mg kg^−1^) (Table 1). Excessive amounts of calcium, whose range cannot be reached (according to data from reference publications) in waste cheese [42], cause the precipitation of carbon and phosphate and the formation of mill scale in reactors. As far as the value obtained in this study is concerned, calcium will only favor the formation of cell aggregates.

There is a relatively high concentration of sodium in cheese and cheese waste, including quark, due to their salt content [3,4]. Sodium in the anaerobic digestion system may easily affect the activity of microorganisms and disturb their metabolism. At low concentrations, that is, 230–350 mg L^−1^, sodium is necessary for methanogens to form adenosine triphosphate and oxidize NADH [42]. Potential problems arising from a slightly elevated sodium level in CE (Table 1) and the acidic pH of this substrate (pH = 4.67) can be solved by using confectionery waste as a co-substrate (neutralizing and complementary composition), buffering sewage sludge and the lignin microbial carrier.

The physicochemical properties of waste wafers (WF) and digested sewage sludge (inoculum) were characterized in detail by the first author in earlier publications [14,15,25,26]. Generally, WF are characterized by approximately neutral pH, considerable content of total solids (TS) and volatile solids (VS), and a high C/N ratio (see Table 1), which contributes to their high biochemical methane potential (BMP) [15]. The concentration of ammonium nitrogen (N–NH_4_^+^) and light metals does not exceed the critical values [39] for the stability of anaerobic digestion (AD). On the contrary, it favors the development and activity of methanogenic consortia [14,26]. The main advantage of stabilized sewage sludge is its high buffer capacity due to its considerable alkalinity developed in the raw sewage sludge digestion process, where nitrates were reduced to carbonates and hydrogencarbonates (total denitrification) [43].

### 3.2. Carrier Characteristics

At present, scientists in many centers all over the world are conducting numerous studies on the physicochemical and structural properties of lignin [44,45,46]. Pilarska et al. (2018) also presented the results of a wide range of analyses in another article on the use of lignin and lignin combined with PVP as a cell carrier [25]. 

Spectroscopic analysis proved that the combination of lignin with PVP had a physical nature and affected its surface properties rather than its chemical ones. The absence of chemical bonds between the carrier and the modifier should be explained by the non-stoichiometric ratio of both compounds. However, surface effect was the only role of the modifier. 

Observations with SEM images showed that both materials tended to form irregularly shaped particle aggregates with a clearly mesoporous structure [25]. Irregular microstructures and troughs favor cell adhesion. At a later stage they favor cell colonization on the surface of the carrier.

As the results showed, the relatively well-developed specific surface area of lignin (1.9 m^2^/g) and lignin+PVP (2.5 m^2^/g), [25] is an indispensable attribute of an effective carrier. Moreover, large pores with diameters up to 20 nm can be seen in Figure 2, which shows the pore size distribution of lignin and lignin+PVP. 

The knowledge of the thermal stability of lignin and lignin+PVP is important because sometimes the AD process takes place under thermophilic conditions. Pilarska et al. (2018) interpreted the results of a thermal gravimetric analysis and differential thermal gravimetric analysis in detail and they proved the high thermal stability of both materials [25]. The first signal of a weight loss of 2.7%–3.9% was recorded only at a temperature of about 200 °C.

Due to various conditions of methane fermentation, including those where adverse changes in the pH of the environment (from acidic to alkaline) may occur, it is recommended to verify the electrokinetic stability of lignin combined with PVP (see Figure 3). The knowledge of the zeta potential value, the parameter which is a measure of electrostatic interactions between particles, enables estimation of the stability of the colloidal dispersion of the system, which is understood as a limited tendency to coagulate particles. In water systems, the limit value of zeta potential of about ±30 mV is assumed to determine the electrokinetic stability of dispersion. When this parameter is equal to zero, the system is in a state of full coagulation. The value at which the zeta potential is equal to zero is called the isoelectric point (IEP) [47].

Kraft lignin, like kraft lignin combined with PVP, exhibited negative zeta potential values (from −40 mV to −25 mV) over the entire pH range under analysis (see Figure 3). The negative surface charge is a consequence of surface ionization of hydroxyl and acid groups, resulting from the dispersion of lignin in aqueous solutions. The distribution of the obtained measurement points proved that both materials exhibited high electrokinetic stability within a wide pH range, that is, from 4 to 11. This observation is in line with data provided in relevant publications [24,45]. The shape of electrokinetic curves suggests that both materials tend to achieve IEP at a pH close to 1 [45].

### 3.3. Batch Experiments

#### 3.3.1. Digestion Process Monitoring

The monitoring of the process stability involved measuring pH and the VFA/TA ratio (volatile fatty acids-to-total alkalinity ratio). As results from the publications, the optimal pH range for the growth of methanogens is 6.5–7.2 [15,42,48]. In none of the samples pH dropped to 6.5, which indicates the accumulation of volatile fatty acids and acidification of the environment (Figure 4). However, an increase in pH to a maximum of 7.55 was noted, especially in the samples containing cheese waste. The increase in pH resulted from the decomposition of milk protein—casein, which was accompanied by the release of ammonia [3]. The equations below apply to hydrolysis (1) and acidogenic phase (2):n-protein-C-NH_2_ + H_2_O → C_x_H_y_O_z_N_a_S_b_ + cP(1)
2 C_x_H_y_O_z_N_a_S_b_ + 5 H_2_O → 2 C_x_H_y_O_z_ + 2a NH_3_ + 2b H_2_S (2)

Apart from the VFA concentration, the volatile fatty acids/total alkalinity (VFA/TA) ratio is commonly used to assess the process stability. The curves corresponding to variation in the VFA/TA ratio during the decomposition of substances in individual samples show that the process proceeded without inhibition, according to the assumed ranges of values specifying stable digester and some instability [15,43] (Figure 4). The highest VFA/TA ratio of 4.9 was noted for the combination of the co-substrates with lignin (WF+CE/lignin). However, after 7 days it dropped to 0.4, which is the process stability limit value. The VFA/TA ratio shows that the decomposition of both CE and WF+CE was particularly stable. These results are correlated with the chemical oxygen demand (COD_r_) values, which began to decrease on the fourth (CE, CE/lignin, WF+CE) or eighth day (WF/lignin, WF+CE/lignin, inoculum, inoculum/lignin), depending on the sample (Figure 5).

The effective removal of VS from the system was confirmed by a decrease in the amount of oxygen consumed by oxidation reactions, mainly organic matter. The best removal efficiency of the total COD_Cr_, that is, 71.5%, was noted for the CE. It was 42.4% for the CE/lignin, and a comparable value, that is, 42.8%, for the WF. There were similar results of experiments where whey cheese was a substrate [9,10].

The decomposition of organic matter also releases nitrogen compounds. The analysis of changes in nitrogen forms occurring during anaerobic digestion is particularly important when protein substrates and sewage sludge are used. In the process of anaerobic decomposition of organic substances, organic nitrogen is converted into ammonium nitrogen, whereas part of organic nitrogen is bound in biomass [49]. The efficiency of ammonium nitrogen formation depends on the digestion chamber load and the process temperature [50]. Snell (1943) was the first to observe that during the decomposition of proteins, 75%–90% of organic nitrogen was converted into ammonium nitrogen [51].

It is justified to make a simultaneous analysis of the total Kjeldahl nitrogen (TKN) content. Its value is the sum of ammonium nitrogen and organic nitrogen, and separately, ammonium nitrogen (N–NH4+), in the decomposed material (see Figure 6 and Figure 7).

The TKN concentration increased systematically in all the systems at the phases of dynamic decomposition of organic matter contained in the samples (Figure 6). The results of the analysis of the N–NH4+ content in the samples showed that its concentration had increased as a result of the conversion of organic nitrogen into ammonium nitrogen (Figure 7). The maximum N–NH4+ content in the TKN in the digestates with wafers and the ones containing wafers with the lignin–PVP carrier was 83%. The content of ammonium nitrogen in TKN in the samples containing only waste cheese was as high as 92%. This value proved that organic nitrogen had been effectively converted and it was in agreement with data provided in reference publications [48]. The analysis of the ammonium nitrogen concentration in the sample with the WF+CE co-substrates per TKN concentration revealed an even higher value at the last phase of the process—98%. Therefore, the addition of the wafer co-substrate improved the degradation. According to Chen et al. (2013), the concentration of N–NH4+ produced during the experiment did not threaten the activity or development of microorganisms [42]. The general distribution of the values was consistent with the results reported by other authors [52].

#### 3.3.2. Bacterial Count and Enzymatic Activity

The results of the research on the influence of three types of organic waste (including the inoculum) and the lignin carrier on the total count of bacteria and the dehydrogenase activity were analyzed statistically. The two-way analysis of variance showed that the factors under analysis had highly significant influence (*p* = 0.001) on the dynamics of variation in the count and activity of bacteria. The multiple regression analysis showed that the second degree model was the best for both the cell count and microbial activity—the coefficient of determination R^2^ for the square model had the highest values (see Figure 8 and Figure 9).

The results of the experiment showed that during the process, the highest count of eubacteria (Figure 8 and Figure 10) was found in the WF+CE/lignin sample, which consisted of waste wafers co-digested anaerobically with waste cheese and the lignin+PVP carrier. It is most likely that this phenomenon was caused by the pH value (pH ~7), which remained the most favorable for bacterial growth and development during the decomposition of material in this variant (Figure 4). This fact was also confirmed by the relatively high value of Pearson’s linear correlation coefficient (see Table 3).

The waste cheese fermented with the CE/lignin carrier was another sample with a high bacterial count (Figure 8). There was a positive correlation between the count of bacteria and the content of ammonium nitrogen (N–NH_4_^+^) in the digestate (Table 3). Ammonium nitrogen is significant for the course of anaerobic digestion. On the one hand, it is the nutrient substance of anaerobic microorganisms. On the other hand, it ensures partial alkalinity of the anaerobic digestion system [53]. As mentioned before, N–NH_4_^+^ has a beneficial effect on condition its concentration does not exceed the value causing the inhibition of the AD process [42].

The results of the microbiological tests showed intensified influence of the type of waste used (and its combinations) with or without the carrier on the count of bacteria in the following gradation: WF+CE ˃ CE ˃ inoculum ≥ WF.

The analysis of the influence of the carrier alone on the proliferation of bacteria showed that the cell count in the experimental variants enriched with lignin was 31%–46% higher. This finding was in line with the results of earlier studies conducted by the same authors [25,26]. The beneficial effect of lignin results from its sorption properties, which extend the activity and stability of the consortium of microorganisms by protecting them from changes in pH, temperature, and composition of the substrate [23,54]. The activating effect of the lignin+PVP carrier could also be attributed to its specific irregular microstructure, ensuring stable contact between lignin and cells. It is said that the cell binding to lignin has covalent nature. Lalov et al. [55] made the same hypothesis when they applied granulated polymeric support (poly(acrylonitrile-acryloamide)). As far as lignin is concerned, it will most likely be the bond formed by the combination of hydroxyl and hydroxymethyl groups from the lignin and amino groups from the cells. This statement will be verified in another study by Pilarska et al.

The experimental combination type was an important parameter influencing the count of methane microorganisms of the *Archaea* domain (Figure 11). The microbiological analysis of the fermented materials made at the last term showed the highest statistically significant count of methane bacteria in the WF/lignin sample. Apart from that, lignin had no statistically significant effect on the count of bacteria in most of the experimental variants.

Figure 9 shows the results of biochemical analyses. Like in the case of the bacterial count, the highest dehydrogenase activity level was found in the WF+CE/lignin and CE/lignin samples. The DHA, which is considered a very important indicator of the intensity of respiratory metabolism, mainly in active microorganisms, is positively correlated with the content of organic matter and with the count of microorganisms [56]. However, the results of our research showed that the DHA was negatively correlated with the bacterial count (Table 3). In most of the variants, the DHA was increasing successively until the 14th day of the experiment. Then, the value of this parameter began to decrease. This phenomenon is most likely to have been caused by the secretion of specific process inhibitors, such as hydrogen sulphide, during the biodegradation of matter [3,53]. The maximum DHA was observed on the 18th day of the experiment, especially in the aforementioned variants: WF+CE/lignin and CE/lignin.

The biochemical analysis of the fermented waste showed that lignin positively affected the DHA in all the experimental variants. This fact proves that carriers stabilize the structure of enzymatic proteins [17].

#### 3.3.3. Process Performance

Table 4 shows the amount of biogas obtained from the waste cheese (quark) (CE) calculated per organic substance, that is, 881.26 m^3^ Mg^−1^ VS, where the share of methane was 556.42 m^3^ Mg^−1^ VS. This is a very good result, comparable with the yield from buttermilk and much better than the yield from whey due to the higher TS and differences in the chemical composition, including a considerable content of fat [1,3,16]. The efficiency of anaerobic digestion of CE is also comparable with that of food waste from restaurants and canteens [57]. The efficiency of biogas production from confectionery waste or stuffed wafers alone was verified and discussed in detail in earlier publications by Pilarska et al. (2018, 2019) [14,15].

The co-substrate system with waste wafers and waste cheese was tested for the first time. The combination of WF with CE gave very good results: 1102.00 m^3^ Mg^−1^ VS of biogas with almost 52% of methane (571.57 m^3^ Mg^−1^ VS). Figure 12 illustrates a successive increase in the accumulated biogas—the process was not inhibited. The analyses of pH, the VFA/TA ratio, TKN, and N–NH_4_^+^ showed that the matter contained in the WF+CE sample was decomposed stably and effectively (see Figure 4, Figure 6 and Figure 7). The results proved the occurrence of synergistic effects between these two materials. There is a good prognosis for the implementation of this solution in industrial production. The analysis and comparison of the earlier results show that dairy waste proved to be a better co-substrate for waste wafers than sewage sludge [14,25].

The addition of lignin to the WF+CE system aided the microorganisms catalyzing the bioconversion process and increased the yield of biogas and methane by 15% (Table 4). This phenomenon was correlated with the results of microbiological and biochemical analyses, which showed that the combining of WF and CE as co-substrates resulted in the highest count of bacteria and the highest dehydrogenase activity of all the experimental variants.

The research showed that due to the properties of lignin, it could be used as a cell carrier in the AD process [25,26]. So far, other researchers have mainly focused on assessing the activation and stabilization of specific enzymes by lignin. For example, lignin was proved to activate α-amylase and lipase [58], where lipase could be immobilized on cellulose/lignin beads [59] or on the chitin/lignin composite [23]. Gong et al. (2017) used lignin extracted from bamboo shoot shells (BBS) to activate and α-amylase immobilization [54]. The authors attributed the active effect of lignin to its rough and relatively well-developed surface, where numerous cavities ensure optimal distribution and stable contact between starch and the enzyme. Researchers observed that lignin-immobilized enzymes were characterized by higher catalytic efficiency and storage stability than their free counterparts [51,58,59]. The observations made in the earlier studies [25,26] and the results of the current microbiological and biochemical analyses lead to a similar conclusion.

## 4. Conclusions

The authors once again confirmed that lignin combined with PVP had beneficial effect on the count of microorganisms and enzymatic activity. This study supplements the results of physicochemical analyses of lignin+PVP. They confirmed its mesoporous structure and high electrokinetic stability within a wide pH range from 4 to 11.

The combined cheese and confectionery waste gave very successful results. The compositions of WF and CE waste complement each other and give synergistic effects. The buffering alkaline pH resulting from the release of N–NH_4_^+^ during the AD of cheese waste may prevent acidification, which accompanies the breakdown of carbohydrates. The course of the decomposition process was stable in all the samples, not only due to the selection of the right substrates, but also due to the stable sewage sludge. At the last phases of the process, the share of N–NH_4_^+^ in TKN exceeded 90%, especially in the samples with the CE. This fact confirmed the conversion of organic nitrogen to ammonium nitrogen during the process and effective degradation of organic matter (especially in the WF+CE and CE samples).

The lignin+PVP cell carrier intensified favorable conditions in the anaerobic bioreactor and improved the biogas efficiency. The biochemical analyses showed that eubacteria proliferated intensively and the dehydrogenase activity increased in the samples containing the carrier, especially in the combination with two co-substrates (WF+CE/lignin) and in the waste cheese sample (CE/lignin). Interestingly, there was a positive correlation between the count of bacteria and the content of ammonium nitrogen (N–NH_4_^+^) produced during the decomposition of the CE/lignin material. The biogas production grew from 1102.00 to 1257.38 m^3^ Mg^−1^ VS (including 52.26% CH_4_) in the WF+CE/lignin combination, and from 881.26 m^3^ Mg^−1^ VS to 989.65 m^3^ Mg^−1^ VS (including 64.20% CH_4_) in the CE/lignin combination.

It is recommended to continue research to confirm the nature of the interactions between cells and lignin. Apart from that, it is necessary to supplement research on the removal of heavy metals from fermented sewage sludge and tests on the absorption of other chemicals contained in potential substrates, which inhibit the AD process.

## Figures and Tables

**Figure 1 polymers-11-02073-f001:**
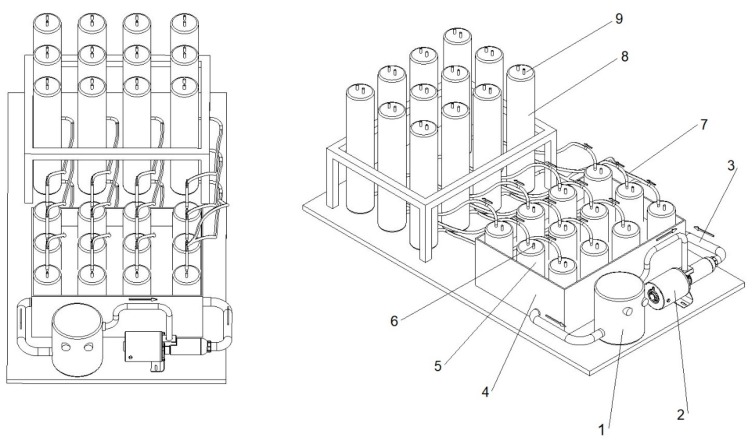
The anaerobic bioreactor (12-chamber section) used in biogas production experiment: 1—water heater; 2—water pump; 3—insulated tubes for heating medium; 4—water jacket (39 °C); 5—bioreactor (1.4 L); 6—slurry sampling valve; 7—tube for biogas transport; 8—graduated tank for biogas; 9—gas sampling valve.

**Figure 2 polymers-11-02073-f002:**
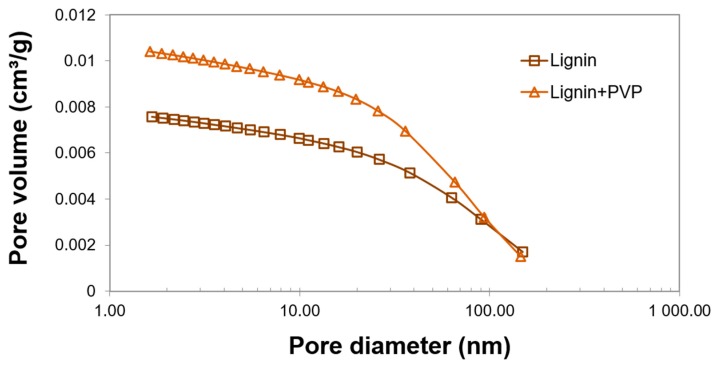
Pore size distribution of the pure and grafted lignin.

**Figure 3 polymers-11-02073-f003:**
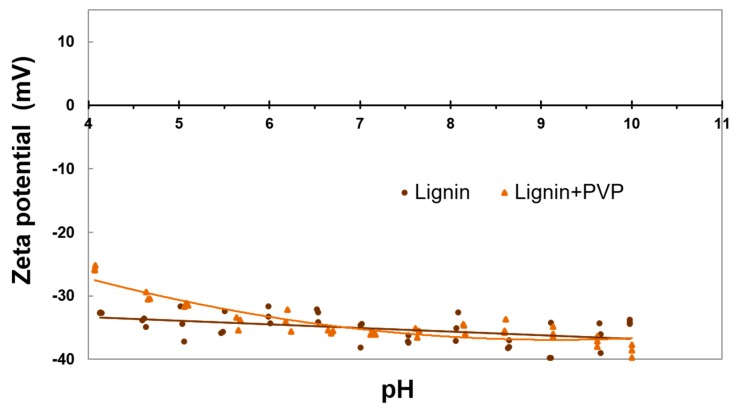
Electrokinetic stability of the pure and grafted lignin.

**Figure 4 polymers-11-02073-f004:**
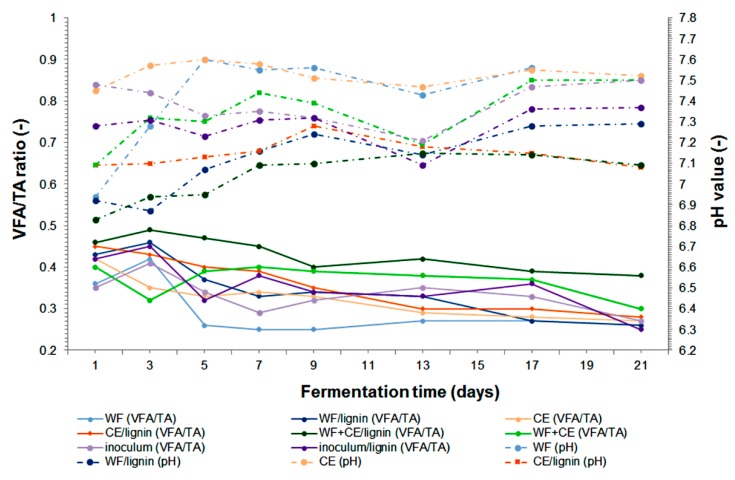
Changes in the pH and VFA/TA ratio during the anaerobic digestion of the samples under analysis. TA: total alkalinity.

**Figure 5 polymers-11-02073-f005:**
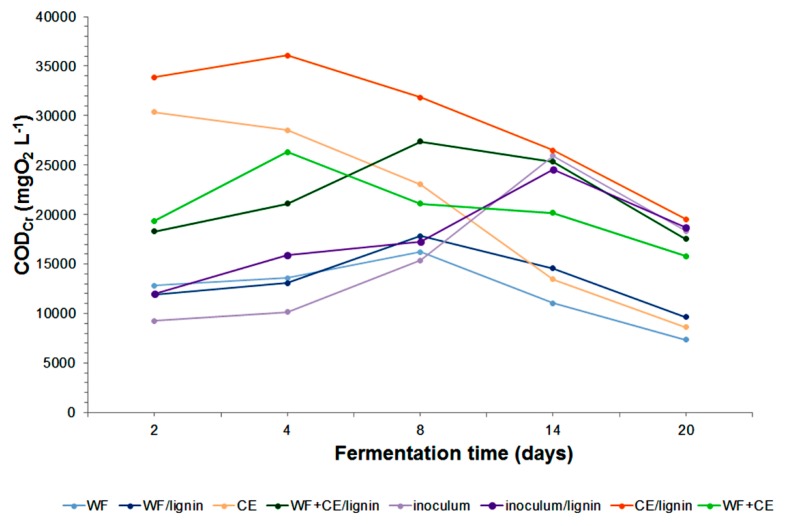
Changes in the COD_Cr_ during anaerobic digestion of the samples tested.

**Figure 6 polymers-11-02073-f006:**
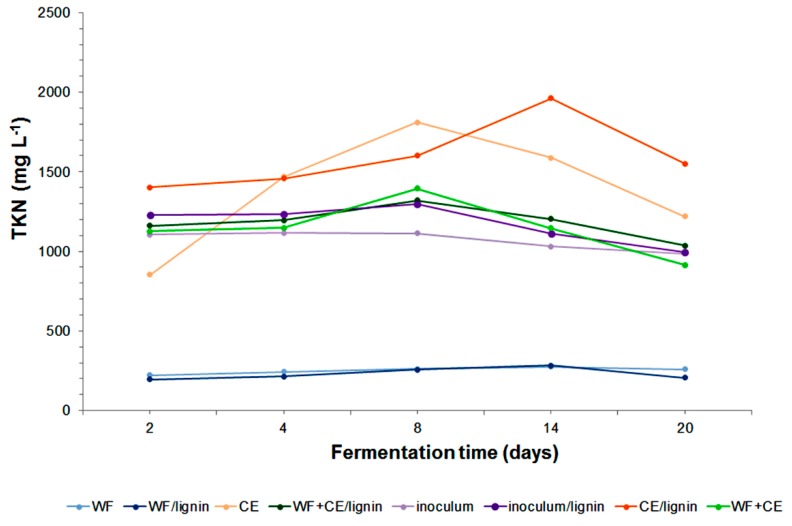
Changes in the TKN (total Kjeldahl nitrogen) during anaerobic digestion of the samples tested.

**Figure 7 polymers-11-02073-f007:**
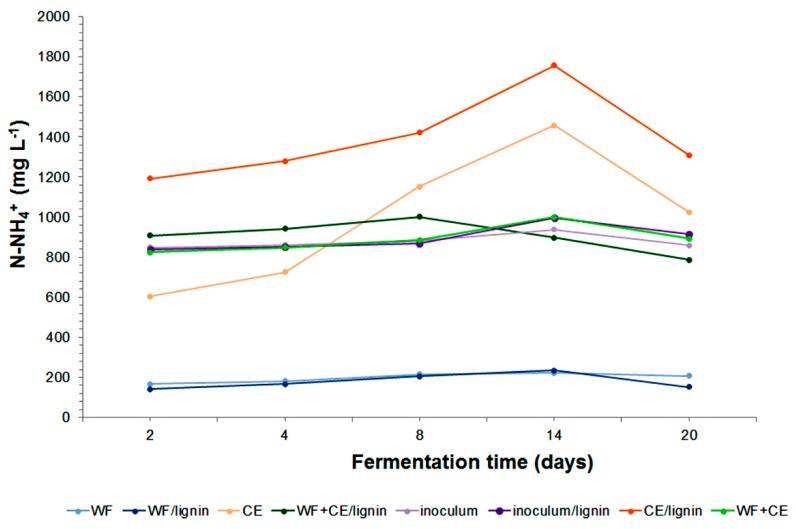
Variation in the N–NH4+ during anaerobic digestion of the samples tested.

**Figure 8 polymers-11-02073-f008:**
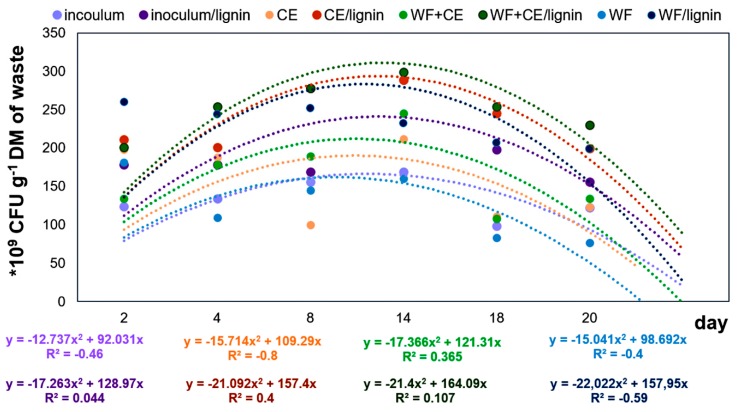
Changes in the total count of bacteria in the samples digested.

**Figure 9 polymers-11-02073-f009:**
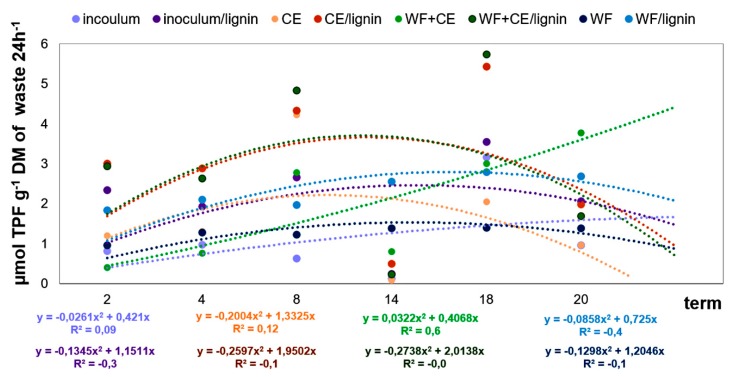
Changes in the dehydrogenase activity (DHA) in the samples digested.

**Figure 10 polymers-11-02073-f010:**
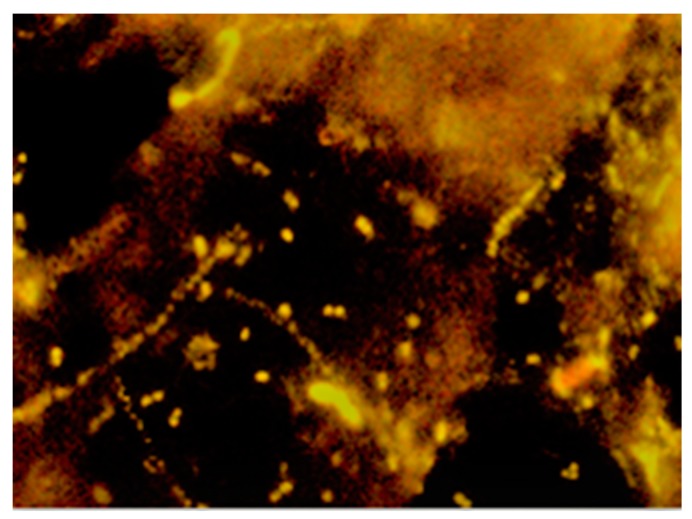
Specific identification of whole fixed bacterial cells with fluorescent oligonucleotide probes (FISH)—WF + CE/lignin combination (400x magnification).

**Figure 11 polymers-11-02073-f011:**
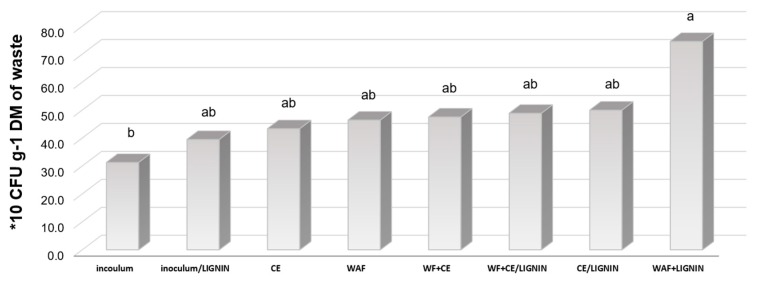
The count of *Archaea* in the samples subjected to anaerobic digestion at the last term of analyses.

**Figure 12 polymers-11-02073-f012:**
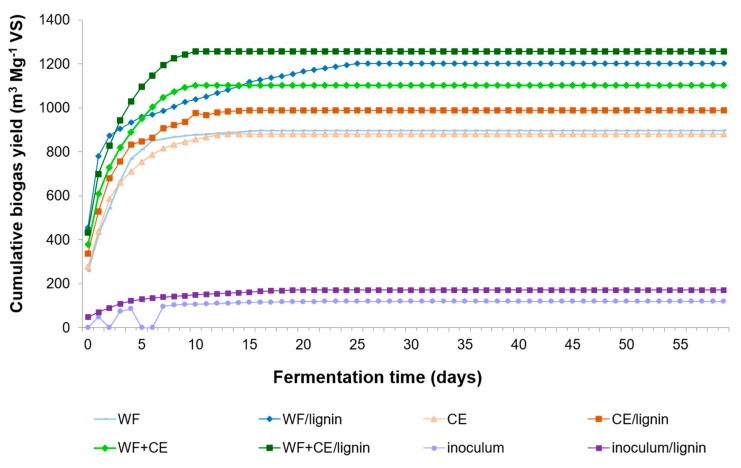
Cumulative biogas production curves from VS of the samples tested.

**Table 1 polymers-11-02073-t001:** Physicochemical properties of substrates and inoculum used in the experiment.

Waste	pH	TS	VS	C/N Ratio	C	N	N–NH_4_	P	Light Metal Ions			
									K	Na	Mg	Ca
	–	(wt %)	(wt %_TS_)	–	(wt %_TS_)	(wt %_TS_)	(wt %_TS_)	(mg kg^−1^)	(mg kg^−1^)			
CE	4.67	24.89	95.75	3.48	52.19	15.01	0.44	1,452	351	433	84	849
WF	6.80	82.24	98.37	46.21	41.59	0.90	0.29	152	37	154	49	154
Inoc.	7.19	2.15	58.33	3.33	27.68	8.32	3.86	2560	3236	6300	33	52

TS = total solids. VS = volatile solids Inoc. = Inoculum.

**Table 2 polymers-11-02073-t002:** Composition and selected properties of the substrate/inoculum batches.

Batch	CE (g)	WF (g)	Lignin+PVP (g)	Inoculum (g)	pH	TS (%)	VS (%)
WF	–	14.0	–	1186.0	7.24	4.24	70.15
WF/lignin	–	14.0	23.5	1186.0	7.16	4.07	67.34
CE	30.7	–	–	1169.9	7.38	2.73	59.29
CE/lignin	30.7	–	23.5	1170.3	7.04	2.76	60.47
WF+CE	4.1	7.8	–	1189.0	7.02	2.74	58.72
WF+CE/lignin	4.1	7.8	23.5	1169.9	6.78	2.75	58.84
inoculum	–	–	–	1200.0	7.22	3.17	69.82
inoculum/lignin	–	–	23.5	1200.0	7.13	4.85	71.01

**Table 3 polymers-11-02073-t003:** Pearson correlation coefficient between bacteria number, DHA activity, and chemical parameters of samples applied in the experiment at consecutive terms of analysis.

Parameters	WF	WF/Lignin	CE	CE/Lignin	WF+CE	WF+CE/Lignin	Inoculum	Inoculum/Lignin
**Bacteria Number**								
N–NH_4_^+^	−0.19	0.21	−0.13	0.81 *	0.67	0.31	0.81 *	0.80 *
TKN	−0.19	0.24	−0.23	0.79	0.60	0.47	0.41	−0.14
pH	−0.46	−0.69	−0.44	0.98 *	−0.42	0.74	−0.88 *	−0.92 *
DHA	−0.70	−0.93 *	−0.73	−0.03	−0.33	−0.28	−0.85 *	−0.70
**DHA**								
N–NH_4_^+^	0.75	0.03	−0.03	−0.56	0.08	−0.23	−0.49	−0.57
TKN	0.79	0.01	0.44	−0.56	−0.04	−0.26	−0.46	−0.08
pH	0.86 *	0.67	0.24	−0.22	0.79	0.13	0.54	0.86 *

Explanation: *—the correlation between the tested parameters is statistically significant with probability *p =* 0.05; DHA—dehydrogenase activity, TKN—Total Kjeldahl Nitrogen.

**Table 4 polymers-11-02073-t004:** Total biogas and methane performance.

Batch	Biogas		Methane		CH_4_ (%)
	(m^3^ Mg^−1^ FM)	(m^3^ Mg^−1^ VS)	(m^3^ Mg^−1^ FM)	(m^3^ Mg^−1^ VS)	
**WF**	866.33	897.22	479.16	496.39	55.31
WF/lignin	966.28	1201.45	551.34	685.53	57.06
CE	210.04	881.26	132.61	556.42	63.14
CE/lignin	235.87	989.65	151.43	635.36	64.20
WF+CE	674.48	1102.00	349.83	571.57	51.87
WF+CE/lignin	769.58	1257.38	402.18	657.68	52.26
inoculum	1.51	120.07	0.94	74.90	62.38
inoculum/lignin	1.29	170.43	2.14	103.10	60.49

FM: fresh matter.

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
