# Peer review of "The Use of Lignin as a Microbial Carrier in the Co-Digestion of Cheese and Wafer Waste"

_polymers, 2019, doi:10.3390/polym11122073_

Round 1

Reviewer 1 Report

Using lignin as a protein or microbial carrier is not new. The authors should focus on how does the lignin preparation used in this study interact with microbial so that the improved or better results could be obtained for the co-digestion. At least, detailed characterization of the lignin used are needed, and why lignin was the key for such a process.

Author Response

Dear Reviewer 1,

thank you very much for your valuable comments. I have made all changes as you recommended. This certainly improved and strengthened the substantive aspects of my study.

I agree that using lignin as a protein or microbial carrier is not new. However, as I wrote in my earlier two publications, it is rarely applied in the anaerobic digested process. My earlier publications also provided a detailed explanation why I had chosen lignin as the microbiological carrier for the AD process. I also made a detailed comparative analysis of lignin and the lignin/PVP carrier (Pilarska et al. Energies, 2018). In this study I wanted to refer to the results and knowledge acquired in earlier research to confirm the beneficial effect of lignin in another system of co-substrates. As this system is very beneficial, it may be really interesting to the reader.

However, I understand that due to the need to maintain the substantive consistency of the article and the profile of the journal it is necessary to make a special reference to the biopolymer carrier used in the research. In fact, not every reader will search for previous, cited publications. I fully agree with this. Thank you very much for your comments.

The changes in the text are marked in yellow.

Kind regards,

Agnieszka Pilarska

Reviewer 2 Report

I have reviewed article entitled “The use of lignin as microbial carrier in the co-digestion of cheese and wafer waste” This is straightforward experimental study which falls within the interest of this journal. I am sure some of the readers will be interested in data presented in this work. I have some comments, questions and suggestions below, once the manuscript is revised based on them I guess it will be ready to be published.

Line 15 Independent just use individual

Line 16 how do you know this is the first study ?? Maybe somebody did and has not been published . This is very challenging statement, you can’t such thing in a scientific paper.

Line 50 what process???

Line 64 another challenging statement ??

Never use “the authors of this….” Sounds funny, Use passive voice throughout the manuscript.

Line 219 one more challenging statement???

Line 270 revise this

Figure 4-10 are not clear hard to see , they need some revision.

Author Response

Dear Reviewer 2,

thank you for your favourable opinion about our article and your valuable and useful comments. I agree with all of them. I have changed my article as recommended.

The expression has been changed as suggested. All challenging statements have been removed from the article. The name of the process has been specified in the suggested place. The Passive Voice has been used in the manuscript (the phrase ‘the authors of this ....’ has been replaced) I checked the excerpt about the zeta potential. The values have been corrected, but the content was correct and in line with the data from the reference publications cited, so no change was necessary. I have improved the legibility and resolution of the charts as indicated.

The changes in the text are marked in yellow.

Kind regards,

Agnieszka Pilarska

Reviewer 3 Report

The manuscript reported on "The Use of Lignin as a Microbial Carrier in the Co-Digestion of Cheese and Wafer Waste". It was generally well written an suitable for publication subject to minor checks and correction. Provide reference support for your methods particularl in section 2.3.1. Also, the absract can be better refined.

Author Response

Dear Reviewer 3,

thank you for your favourable opinion about our article and your valuable comments. I have followed all the reviewers’ advice and changed my article as you recommended. This certainly has improved the quality of my article.

As you suggested, I have improved the abstract and emphasised the role of lignin as a microbial carrier in anaerobic digestion. As you suggested, I have supplemented the reference support in the methodology, especially in section 2.3.1.

The changes in the text are marked in yellow.

Kind regards,

Agnieszka Pilarska